# Integrated psychosocial, economic strengthening, and clinical service-delivery to improve health and resilience of adolescents living with HIV and their caregivers: Findings from a prospective cohort study in Zambia

**Joseph G. Rosen** *, **Lyson Phiri** , **Mwelwa Chibuye, Edith S. Namukonda, Michael T. Mbizvo, Nkomba Kayeyi**

Population Council, Lusaka, Zambia

* joseph.gregory.rosen@gmail.com

**Data Availability Statement:** The data underlying the results presented in this study are available

## Abstract

### Background

Children and youth are profoundly impacted groups in Zambia's HIV epidemic. To evaluate delivery of integrated psychosocial, economic strengthening, and clinical services to HIV-affected households through the Zambia Family (ZAMFAM) Project, a prospective cohort study compared socio-economic, psychosocial, and health outcomes among ZAMFAM beneficiaries to non-beneficiaries.

### Methods

In July–October 2017, 544 adolescents living with HIV (ALHIV) aged 5–17 years and their adult caregivers were recruited from Central (ZAMFAM implementation sites) and Eastern (non-intervention sites) Provinces. Structured interviews at baseline and one-year follow-up assessed household characteristics, socio-economic wellbeing, and health service utilization. Poisson regression with generalized estimating equations measured one-year changes in key health and socio-economic indicators, comparing ZAMFAM beneficiaries to non-beneficiaries.

### Results

Overall, 494 households completed two rounds of assessment (retention rate: 91%) Among ALHIV, improvements in current antiretroviral therapy use over time (Adjusted Prevalence Rate Ratio [aPRR] = 1.06, 95% Confidence Interval [95% CI]: 1.01–1.11) and reductions in non-household labor (aPRR = 0.44, 95% CI: 0.20–0.99) were significantly larger among ZAMFAM beneficiaries than non-beneficiaries. For caregivers, receiving ZAMFAM services was associated with significant reductions in HIV-related stigma (aPRR = 0.49, 95% CI: 0.28–0.88) and perceived negative community attitudes towards HIV (aPRR = 0.77, 95% CI: 0.62–0.96). Improvements in caregiver capacity to pay for unexpected (aPRR = 1.54,

**Funding:** This study was supported by Project SOAR (AID-OAA-A-14-00060), funded by the U.S. President's Emergency Plan for AIDS Relief (PEPFAR) and the U.S. Agency for International Development (USAID). The funders played no role in study design, data collection and analysis, decision to publish, or preparation of the manuscript. The contents of this publication are the sole responsibility of the authors and do not necessarily reflect the views of USAID, PEPFAR, or the United States government.

**Competing interests:** The authors have declared that no competing interests exist.

**Abbreviations:** ALHIV, Adolescents living with HIV; ART, Antiretroviral therapy; CHW, Community health worker; DAPP, Development Aid from People to People; ECR, Expanded Church Response; NGO, Non-governmental organization; OVC, Orphaned and vulnerable children; PEPFAR, U.S. President's Emergency Plan for AIDS Relief; USAID, U.S. Agency for International Development; VLS, Viral load screening; ZAMFAM, Zambia Family (*project*).

95% CI: 1.17–2.04) and food-related expenses (aPRR = 1.48, 95% CI: 1.16–1.90), as well as shared decision-making authority in household spending (aPRR = 1.41, 95% CI: 1.04–1.93) and self-reported good or very good health status (aPRR = 1.46, 95% CI: 1.14–1.87), were also significantly larger among ZAMFAM beneficiaries.

## Conclusions

Significant improvements in caregivers' financial capacity were observed among households receiving ZAMFAM services, with few changes in health or wellbeing among ALHIV. Integrated service-delivery approaches like ZAMFAM may yield observable socio-economic improvements in the short-term. Strengthening community-based delivery of psychosocial and health support to ALHIV is encouraged.

## Background

Zambia's National AIDS Strategic Framework (2017–2021) acknowledges the socio-developmental challenges associated with the HIV epidemic [1], embracing the UNAIDS call to achieve the 95–95–95 targets through universal HIV testing and treatment [2]. Despite noteworthy reductions in HIV incidence and improved outcomes along the care continuum, Zambia's generalized HIV epidemic, with an estimated adult prevalence of 12.3% [3], persists unrelentingly. Children and younger adolescents are profoundly affected groups: an estimated 8,000 children 0–14 years and 4,000 younger adolescents 15–19 years (prevalence: 2.5%) in Zambia are living with HIV [3]. Adolescents also fall substantially behind older adults in the HIV care continuum: fewer than half (40.2%) of 15-24-year-olds living with HIV know their HIV status (adults 35–49 years: 71.3%), and fewer than three-fourths (73.8%) of adolescents on HIV treatment are virally suppressed (adults 35–49 years: 89.8%) [3]. While adolescents account for only 9% of new HIV infections and have substantially lower HIV prevalence (1.1%) relative to adults [3, 4], AIDS-related mortality in this age group remains unacceptably high [4]. Over 600,000 children in Zambia have been orphaned as a direct consequence of the HIV epidemic, and an estimated 1.3 million, roughly 10% of Zambia's population, are left vulnerable in the epidemic's wake [5].

A priority population group for HIV services in Zambia and throughout sub-Saharan Africa is HIV-orphaned and vulnerable children (OVC), whose livelihoods are characterized by extreme poverty, protracted sources of adversity, and high disease burden. The prolonged illness or death of a household member, or addition of an orphaned child to the home, can disrupt social and financial stability, heightening vulnerability to malnutrition, social exclusion, educational attrition, and abuse [6–9]. These outcomes further constrain economic and educational opportunities, trapping HIV-affected children and households in a vicious cycle of poverty and poor health. Changes in the primary caregiver or care environment may equally induce distress, depress agency, and diminish self-efficacy—propagating an HIV risk environment characterized by earlier sexual debut, forced sex or sexual coercion, and condomless sex with multiple partners [9–15]. Adolescents living with HIV (ALHIV) are particularly vulnerable to suboptimal care and treatment outcomes because these cumulative adversities (e.g., financial insecurity and caregiver instability) destabilize access to and retention in HIV services [16–18].

Research has demonstrated the positive impact community-based care and/or support programs can have on ALHIV wellbeing. Findings from a randomized assessment of the Suubi-

Maka initiative in rural Uganda highlight the effectiveness of community-based economic empowerment initiatives to build savings capacity of youth, as well as mitigate negative HIV-related attitudes and risks [19]. The community-based Bwafano project in urban Zambia, likewise, has yielded promising educational retention results, specifically a 15.7% increase in schooling progression among OVC [20]. Results from similar initiatives in Ethiopia [21, 22], Uganda [23], Haiti [24], and China [25] have demonstrated the effectiveness of community-based programming on psychosocial, financial, nutritional, and HIV-related outcomes for ALHIV. There is a paucity of literature, nonetheless, describing impacts of multi-component interventions on various health and wellbeing outcomes for both ALHIV and their caregivers, particularly in Zambia. An improved understanding of the mechanisms through which multi-component interventions influence HIV-related morbidity and socio-structural vulnerabilities (e.g., food insecurity, educational retention) is needed to identify optimal and sustainable solutions to cross-cutting health and development challenges.

This paper presents findings from a one-year assessment of one such community-based support initiative, the Zambia Family (ZAMFAM) Project, on psychosocial, socio-economic, and health outcomes among ALHIV and their adult caregivers. This observational, prospective cohort study measures longitudinal changes across multidimensional health and resiliency indicators, comparing program beneficiary to non-intervention households. Findings contribute to a growing body of evidence underscoring the effectiveness of community-based care and support programs for ALHIV and their caregivers.

## Materials and methods

### The Zambia Family (ZAMFAM) Project

The ZAMFAM Project, launched in 2015 with support from the United States Agency for International Development (USAID) via the U.S. President's Emergency Plan for AIDS Relief (PEPFAR), employs a community-based, integrated service delivery model to strengthen household capacity in meeting the needs of OVC living with or affected by HIV, as well as improve child and caregiver wellbeing [26]. Dovetailing existing nationally scaled interventions for OVC and ALHIV in Zambia, ZAMFAM's intervention components tap into clinical and community volunteer networks to improve case management and outcomes for both OVC and ALHIV, from schooling to retention in HIV care. Recognizing the role of households and communities in shaping OVC and ALHIV livelihoods, ZAMFAM also engages primary caregivers in parenting and economic strengthening activities (e.g., village loans groups), aiming to meet their psychosocial, financial, and nutritional needs in order to foster more optimal, resilient care environments for their dependents. ALHIV have been prioritized for program support in the four provinces where ZAMFAM activities are delivered by two implementing partners: Development Aid from People to People (DAPP), implementing in Central and Southern Provinces, and the Expanded Church Response (ECR) Trust, implementing in Copperbelt and Lusaka Provinces.

Working primarily through government structures that oversee activities for vulnerable populations, cadres of para-social workers and volunteer caregivers conduct regular household visits to develop family-tailored plans and motivate uptake of available community-level activities. These activities include needs-based, age-appropriate interventions addressing multiple dimensions of adversity impacting households' ability to meet the needs of OVC and ALHIV. By providing households with the resources and skills to meet unrelenting financial and social challenges that can traditionally destabilize health-seeking behaviors and healthcare engagement, ZAMFAM interventions seek to foster resilience among ALHIV and caregivers, so they can continue prioritizing their health needs in the face of overlapping adversities.

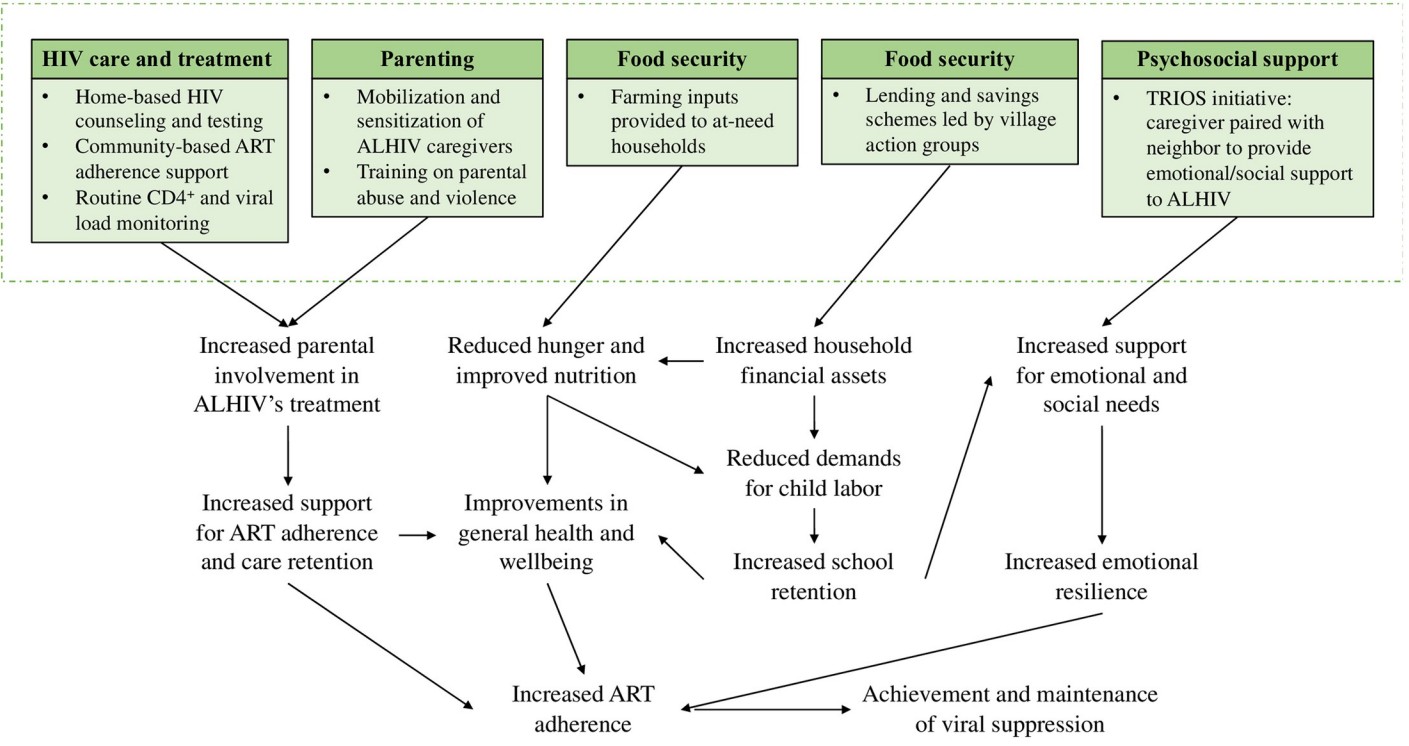

**Fig 1. Intervention impact model for the Zambia Family (ZAMFAM) Project.**

To address these multilevel sources of adversity, the ZAMFAM package of interventions cut across five key domains, which are illustrated in Fig 1 and described in detail below:

- **HIV care and treatment**. Program implementers convene health staff and community health workers (CHWs), identifying strategies for scaling HIV testing (i.e., enhanced home-based HIV counseling and testing, index case testing and partner notification services) as well as viral load screening (VLS) and CD4+ count monitoring for those living with HIV. CHWs are additionally paired with ALHIV to monitor and support adherence to care and treatment.

- **Parenting**. Contracting with a local women's faith-based organization, program implementers train community staff on parenting skills, including child abuse and gender-based violence. Community staff subsequently mobilize caregivers of ALHIV and hold meetings to discuss these issues.

- **Food security**. Farming inputs—including maize seed, legumes, cassava, sweet potatoes, chickens, and goats—are provided to OVC and ALHIV households.

- **Household economic strengthening**. Loans and savings schemes are introduced into Village Action Groups, allowing for members to borrow money for investments (e.g., in businesses, essential household items) or purchase agricultural inputs at lower interest rates.

- **Psychosocial support**. ALHIV are paired with their primary caregiver and an adult from a neighboring household to support the adolescent's adherence to ART and engagement with HIV services. Counselors and CHWs additionally receive specialized training to enhance their counseling skills.

ZAMFAM's multilevel interventions supplement an already comprehensive package of health and social support services for OVC, ALHIV, and their households. In 2016, Zambia adopted a 'Test and Start' strategy, whereby newly diagnosed HIV-positive persons, irrespective of CD4+ cell count or clinical stage, are offered antiretroviral treatment (ART) [27]. VLS has been rapidly scaled up, with recommendations to provide routine (every 6 months) viral load monitoring [27]. Counseling and health education are almost universally available at health facilities, with a smaller number providing nutritional support to qualified households. Non-governmental organizations (NGOs) have coordinated with government facilities and communities to form support groups for people living with HIV, including for youth. Lastly, some agricultural inputs are made available to qualified farmers, most of whom practice subsistence farming.

## Study sites

To appraise ZAMFAM's impact on beneficiaries and households, ALHIV and their primary caregivers in Central (DAPP implementation sites) and Eastern (non-implementation comparison sites) Provinces of Zambia were recruited into a prospective cohort study. In Central Province, participants were recruited from four PEPFAR priority districts—Chibombo, Kabwe, Kapiri Mposhi, and Mumbwa. Four matched districts in Eastern Province—Nyimba, Chipata, Petauke, and Lundazi—were selected as non-intervention, comparison sites based on HIV prevalence estimates for children under 15 from 2016–17 Health Management Information System records of new HIV infections in selected districts as well as the districts' geographic positioning along main thoroughfares. Districts in Eastern Province were selected, in part, due to their distance from ZAMFAM implementation sites in Central Province, in attempt to mitigate potential spillover effects from intervention sites. Importantly, comparison sites in Eastern Province did not have similar or comparable multi-component interventions to ZAMFAM at the time of study enrollment, as determined by key informant interviews and planning meetings with provincial and district health officers in the Ministry of Health.

## Sampling

In Central Province, a two-stage sampling procedure, stratified by urban and rural residence, was implemented to select wards proportional to the estimated ALHIV population size, and a fixed number of households were subsequently sampled in each ward. Smaller wards were combined to form an aggregate ward when households with ALHIV were too few. Sampling proportional to population size allowed for households in wards with large numbers of eligible ALHIV to have a higher probability of enumeration, while selecting a fixed number of households in each ward allowed for the sample to be self-weighting at the analysis stage [28]. Following random selection of 13 wards, pre-consenting beneficiary households with ALHIV aged 5–17 years were organized chronologically by ZAMFAM beneficiary identification codes and were randomly selected for recruitment. A total of 320 households were approached and pre-consented during study recruitment.

In Eastern Province, registers from public health facilities and NGOs providing HIV testing and treatment services in selected districts were used to compile lists of eligible children whose primary caregivers could be approached for pre-consenting and study recruitment. Since newly diagnosed ALHIV (between May 2016 and October 2017) or those recently enrolled in HIV care were too few to form a sampling frame, all eligible ALHIV and their primary caregiver were recruited into the study until a sample size threshold of 272 caregiver-ALHIV pairs was reached. CHWs and health facility staff facilitated the pre-consenting process.

## Data collection

Trained study enumerators not affiliated with ZAMFAM service-delivery efforts administered surveys to consenting ALHIV-caregiver dyads in the household. Participants were surveyed twice during the study period. At baseline (July–October 2017), selected households providing written informed consent completed a structured, tablet-based survey, programmed in SurveyCTO (Cambridge, MA). Caregivers were interviewed first and answered questions pertaining to household characteristics, social and economic wellbeing, and their own health. Following written parental permission and written child assent, ALHIV aged 10–17 years completed a similar structured interview, responding to questions covering HIV care and treatment experiences, general health and service utilization, psychosocial wellbeing, and nutrition. Information about eligible ALHIV aged 5–9 years was obtained by proxy, through an interview with the child's caregiver.

At endline (July–September 2018), study enumerators returned to households sampled at baseline for a follow-up interview using similar data collection materials. ALHIV who relocated to new households or locales during the study period were eligible for follow-up only if the child's new residence was traceable, even if it was located outside the initial sampling frame. Mortality information was obtained from the caregivers of ALHIV who died in between study visits.

In line with ethical protocols governing participant eligibility, distinct survey modules and administration procedures were used for ALHIV of different age groups to protect safety and wellbeing of ALHIV aged 9 years and younger. Surveys were administered in English or in one (or combination of) three local languages—Bemba, Nyanja, and Tonga.

## Measures

**Outcomes.**   Outcomes assessed in the cohort spanned key health and wellbeing indicators articulated in ZAMFAM's theory of change. In addition to addressing HIV-related outcomes, ZAMFAM's core service package tackles key social and structural factors driving ALHIV's. Indicators of wellbeing, therefore, measure the distal (indirect) mechanisms that ZAMFAM interventions act upon (e.g., social support, depression) to bolster health outcomes in ALHIV. These domains included social protection and psychosocial wellbeing (i.e., having someone to help with chores when sick); education and labor (i.e., consistent past-week school attendance, progressed in school in the previous academic year, engaged in any non-household labor for income); financial security (i.e., caregiver meets child's needs better than other households, joint decision-making with spouse in how earned money is spent, household more financially secure compared to others, ability to access money to pay for food, education, or educational expenses); food and nutrition (i.e., ate a smaller meal than desired, skipped a meal because food was unavailable, went to bed hungry, went a whole day and evening without eating); physical health and wellness (i.e., self-reported health is good or very good, too sick to participate in daily activities at least once in the past month, hospitalized for any illness in the past 6 months); HIV clinical outcomes (i.e., caregiver has ever been tested for HIV, currently on ART, past-month daily ART adherence, past-year retention in HIV treatment, CD4+ cell count monitoring in the past 6 months, and VLS in the past 6 months); and HIV-related knowledge, attitudes, and behaviors (i.e., caregiver supports corporeal punishment in home or school, ALHIV ever discussed sex with a household/family member, ALHIV ever discussed HIV/AIDS with a household/family member). The aforementioned covariates were treated dichotomously in analysis, comparing self-reported affirmative responses for each measure in ALHIV and their caregivers, respectively, to non-affirmative responses. Operationalization procedures for specific outcomes, derived from

**Table 1. Operationalization of key Zambia Family (ZAMFAM) Project outcome variables.**

| Outcome | Definition |
|---|---|
| *Basic social support across four domains** | % who have someone in their life: 1) to turn to for suggestions, 2) who can help with chores, 3) that shows love/affection, *and* 4) to do something enjoyable with. |
| *Depressive symptoms** | % experiencing any one of five of depression symptoms (disengagement, hopelessness, restlessness, lethargy, or irritability) in the past six months. |
| *Stigma and mistreatment** | % reporting any one of eleven experiences of stigma, discrimination, or mistreatment due to their HIV status:<br>• You were treated badly at work or lost your job.<br>• You were treated badly at school or were excluded from school activities.<br>• You have had difficulty finding sexual partners.<br>• Your family did not care for you when you were sick.<br>• You were treated badly by health professionals.<br>• You lost friends.<br>• You were treated badly by family or excluded from family activities.<br>• You experienced a break-up of a relationship.<br>• Your community (village) treated you like a social outcast.<br>• You experienced physical violence.<br>• You were treated badly at church or excluded from religious activities. |
| *Negative community attitudes towards PLHIV** | % affirming the presence of any one of seven negative attitudes in their respective communities towards people living with HIV:<br>• Most people think PLHIV are disgusting.<br>• Most PLHIV are rejected when others learn of their HIV status.<br>• PLHIV lose jobs when employers learn of their HIV status.<br>• PLHIV do not get good healthcare if others know about their HIV status.<br>• Most people are uncomfortable around someone living with HIV.<br>• PLHIV are treated differently than people not living with HIV.<br>• Most people believe a person who has HIV is dirty. |
| *Comprehensive HIV knowledge*$^{\Psi}$ | % answering all the following questions correctly:<br>• Can people reduce their chances of getting the HIV virus by having just one uninfected sex partner who has no other sexual partners?<br>• Can people reduce their chance of getting the virus by using a condom every time they have sex?<br>• Is it possible for a healthy-looking person to have the HIV virus?<br>• Can people get the HIV virus from mosquito bites?<br>• Can people get the HIV virus through supernatural means? |
| *Attitudes towards spousal violence*$^{\Psi}$ | Agrees wife-beating acceptable in at least one of the five following situations:<br>• Wife goes out without telling husband.<br>• Wife is not looking after the children.<br>• Wife argues with husband.<br>• Wife refuses to have sex with husband.<br>• Wife burns the food. |

*Constructs and items were derived from MEASURE Evaluation's OVC Survey Toolkit [29].

$^{\Psi}$ Constructs and items were derived from ICF International's Demographic and Health Surveys [30].

MEASURE Evaluation's OVC Survey Toolkit [29] and the Demographic and Health Surveys [30], are further described in Table 1.

**Independent variables.** Socio-demographic factors assessed among ALHIV included age group (5–9, 10–17 years), sex, current school enrollment (attending school or not for those aged 7 years and older), birth certificate possession (self-reports having a birth certificate or not), and disability status (self-reported physical, intellectual, or psychological impairment).

Caregivers provided similar socio-demographic information, with the addition of marital status (married/cohabiting, never married, separated/widowed/divorced), educational attainment (no formal, primary, secondary or higher), employed full-time for income (reports working full-time for money), and functional literacy (assessed by having caregivers read a sentence aloud in English or preferred local language and explain the meaning).

Household indicators included dichotomized shelter protection, determined through enumerator-observation of adequacy of a home's roofing material and walls to provide shelter; any livestock and agricultural land ownership; and recent (past 12 months) death of any household member.

## Data analysis

Descriptive statistics for ALHIV and caregiver socio-demographic and household factors were calculated and compared across study groups at baseline. Chi-square tests of association (for dichotomous variables) and non-parametric Wilcoxon rank-sum tests (for continuous variables) were implemented to identify statistically significant ($p<0.05$) covariate differences, comparing ZAMFAM beneficiaries to non-beneficiaries. Significant one-year changes in key health and wellbeing outcomes within study groups (i.e., ZAMFAM beneficiaries and non-beneficiaries, respectively) was determined using McNemar's test for paired nominal data.

To compare changes in key health and wellbeing indicators from baseline to endline between ZAMFAM beneficiaries and non-beneficiaries, bivariate and multivariable Poisson regression with generalized estimating equations, robust error variance (for binomial outcomes) [31], and an exchangeable correlation structure modeled prevalence rate ratios (PRRs) for outcomes over the one-year assessment period. Regression coefficients from a time-by-group interaction term were population-averaged estimates, pooling fixed and random effects from repeated clustered observations, quantifying differences in key outcomes between study groups over time. Multivariable models adjusted for socio-demographic and household characteristics with statistically significant ($p<0.05$) differences between study groups at baseline. Data were managed and analyzed in Stata/IC 14.2 (StataCorp LLC, College Station, TX).

## Ethical statement

The research protocol was approved by the Population Council Institutional Review Board (New York, NY, USA) and the local ethical body, ERES Converge (Lusaka, Zambia). Administrative approvals were also received from the National Health Research Authority, Ministry of Health, and the Ministry of Community Development and Social Services. The study also established a Technical Advisory Group, whose membership comprised diverse expertise in research implementation and utilization, service delivery, health and social welfare policy, and civil society advocacy. Adult caregivers provided written consent prior to participation; ALHIV written assent and caregiver written consent was obtained for participants younger than 18 years.

## Results

### Baseline cohort characteristics

Fig 2 illustrates sampling, cohort, and retention statistics for the 544 households enrolled into the cohort at baseline. Among 752 households initially approached for study enrollment (ZAMFAM beneficiaries: $n = 335$, non-beneficiaries: $n = 417$), a total of 544 ALHIV and caregiver dyads (ZAMFAM beneficiaries: $n = 272$, non-beneficiaries: $n = 272$) were enrolled at baseline. At endline, 494 ALHIV and caregivers completed the follow-up assessment, yielding

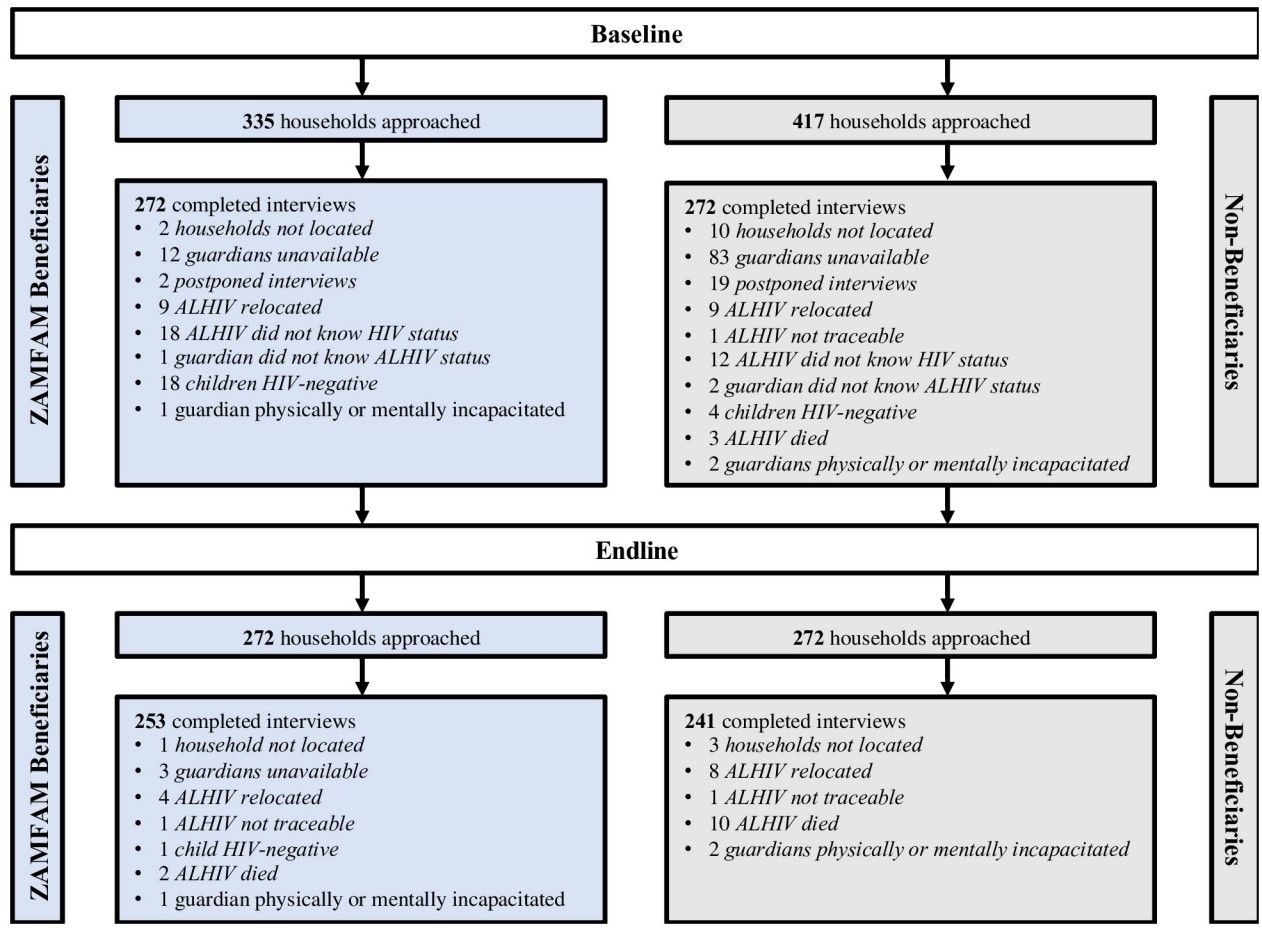

**Fig 2. Recruitment and retention outcomes of the Zambia Family (ZAMFAM) cohort study.**

a 91% retention rate across provinces, with comparable retention estimates recorded among ZAMFAM beneficiaries (93%) and non-beneficiaries (89%). Twelve ALHIV deaths were recorded during the study period, a majority of which occurred among non-beneficiaries ($n = 10$). Eleven of these deaths were attributed to HIV/AIDS-related complications.

Table 2 presents socio-demographic and household characteristics of the baseline sample, stratified by study group. The majority of ALHIV (58.6%) were aged 10–17 years (median: 11 years, IQR: 8–14), and most were female (58.5%). Among ALHIV aged 7 years and older ($n = 468$), three-fourths (76.3%) were enrolled in school at the time of survey—a significantly larger proportion of whom were ZAMFAM beneficiaries than non-beneficiaries (81.3% vs. 71.4%, $p = 0.012$). Fewer than one-fourth (24.1%) had a birth certificate, and 6% had a disability, with twice the number of ALHIV with disabilities in ZAMFAM beneficiary households than in non-beneficiary households (8.1% vs. 4.1%, $p = 0.048$).

Caregivers were predominantly aged 35–54 years (median: 43 years, IQR: 35–53), female (87.9%), married or cohabiting (54.2%), and never formally educated (64.9%). One-fourth (26.8%) were widowed, a significantly higher proportion of whom were ZAMFAM beneficiaries compared to non-beneficiaries (33.1% vs. 20.6%, $p<0.001$). A quarter (27.9%) were employed full-time for money, and nearly half were functionally literate (48.5%). Nearly 9% of caregivers reported a disability, a majority of whom were ZAMFAM beneficiaries (13.6% vs. 4.0%, $p<0.001$). Nearly two-thirds (62.0%) of caregivers were living with HIV.

**Table 2. Socio-demographic and household characteristics of adolescents living with HIV (ALHIV) and their adult caregivers at baseline, by Zambia Family (ZAM-FAM) Project beneficiary status (N = 544).**

| | ZAMFAM Beneficiaries (%) n = 272 | Non-Beneficiaries (%) n = 272 | Total (%) N = 544 | p-value |
|---|---|---|---|---|
| *ALHIV* | | | | |
| Median age, in years (IQR) | 11 (8–13) | 11.5 (8–14) | 11 (8–14) | 0.065 |
| Age group, in years | | | | 0.433 |
| 5–9 | 117 (43.0) | 108 (39.7) | 225 (41.4) | |
| 10–17 | 155 (57.0) | 164 (60.3) | 319 (58.6) | |
| Sex | | | | 1.000 |
| Male | 113 (41.5) | 113 (41.5) | 226 (41.5) | |
| Female | 159 (58.5) | 159 (58.5) | 318 (58.5) | |
| Currently enrolled in school (n = 468) | 187 (81.3) | 170 (71.4) | 357 (76.3) | **0.012** |
| Has a birth certificate | 69 (25.4) | 62 (22.8) | 131 (24.1) | 0.483 |
| Has a disability | 22 (8.1) | 11 (4.1) | 33 (6.1) | **0.048** |
| *Caregivers* | | | | |
| Median age, in years (IQR) (n = 543) | 44 (36.5–55) | 42 (35–50) | 43 (35–53) | **0.027** |
| Age group, in years (n = 543) | | | | **0.046** |
| 18–34 | 51 (18.8) | 66 (24.4) | 117 (21.6) | |
| 35–54 | 151 (55.5) | 157 (57.9) | 308 (56.7) | |
| 55–85 | 70 (25.7) | 48 (17.7) | 118 (21.7) | |
| Sex | | | | 0.431 |
| Male | 36 (13.2) | 30 (11.0) | 66 (12.1) | |
| Female | 236 (86.8) | 242 (89.0) | 478 (87.9) | |
| Marital status | | | | **<0.001** |
| Married/cohabitating | 126 (46.3) | 169 (62.1) | 295 (54.2) | |
| Never married | 15 (5.5) | 6 (2.2) | 21 (3.9) | |
| Divorced/separated | 41 (15.1) | 41 (15.1) | 82 (15.1) | |
| Widowed | 90 (33.1) | 56 (20.6) | 146 (26.8) | |
| Educational attainment | | | | 0.058 |
| No formal education | 176 (64.7) | 177 (65.1) | 353 (64.9) | |
| Primary | 91 (33.5) | 80 (29.4) | 171 (31.4) | |
| Secondary or higher | 5 (1.8) | 15 (5.5) | 20 (3.7) | |
| Employed full-time for income | 83 (30.5) | 69 (25.4) | 152 (27.9) | 0.181 |
| Functionally literate | 138 (50.7) | 126 (46.3) | 264 (48.5) | 0.303 |
| Has a disability | 37 (13.6) | 11 (4.0) | 48 (8.8) | **<0.001** |
| HIV-positive (n = 519) | 154 (59.2) | 168 (64.9) | 322 (62.0) | 0.186 |
| *Household* | | | | |
| Shelter protection | 227 (83.5) | 260 (95.6) | 487 (89.5) | **<0.001** |
| Residence type | | | | **0.301** |
| Rural | 129 (47.4) | 117 (43.0) | 246 (45.2) | |
| Urban | 143 (52.6) | 155 (57.0) | 298 (54.8) | |
| Owns livestock | 43 (15.8) | 83 (30.5) | 126 (23.2) | **<0.001** |
| Owns agricultural land | 148 (54.4) | 182 (66.9) | 330 (60.7) | **0.003** |
| Recent death of HH member | 56 (20.6) | 29 (10.7) | 85 (15.6) | **0.001** |

Significant differences in household characteristics between study groups highlight differential socio-economic vulnerabilities between ZAMFAM beneficiaries and non-beneficiaries at baseline. While nearly 90% of households had shelter protection, a significant 12% difference in shelter protection was recorded between ZAMFAM beneficiaries and non-beneficiaries

(83.5% vs. 95.6%, $p<0.001$). Ownership of livestock (30.5% vs. 15.8%, $p<0.001$) and agricultural land (66.9% vs. 54.4%, $p = 0.003$), respectively, were significantly higher among ZAMFAM beneficiary households than non-beneficiaries. Compared to non-beneficiaries, twice as many ZAMFAM beneficiary households reported a recent (past 12 months) death in the household (20.6% vs. 10.7%, $p = 0.001$).

## One-year differences in key outcomes

**ALHIV.** Among ZAMFAM-beneficiary ALHIV (see Table 3), significant increases in having basic social support across four domains (Δ +8.6%), self-reported good or very good health (Δ +17.5%), current ART use (Δ +6.1%), VLS in the past 6 months (Δ +24.0%) and comprehensive HIV knowledge (Δ +15.7%) were observed from baseline to endline. Significant reductions were recorded in perceived negative community attitudes towards HIV-positive people (Δ –12.9%) and eating a smaller meal than desired at least once in the past month (Δ –8.7%) between survey rounds. Among non-beneficiaries, by comparison, significant improvements in the proportion of ALHIV self-reporting good or very good health status (Δ +18.6%) and VLS (Δ +21.6%) mirrored changes observed for ZAMFAM-beneficiary ALHIV. Significant reductions in non-beneficiary ALHIV who were too sick to participate in daily activities in the past month (Δ –8.1%) were also recorded.

**Caregivers.** Among ZAMFAM-beneficiary caregivers (see Table 4), significant increases in having help with daily chores when sick (Δ +7.6%), joint decision-making with spouse in how earned money spent (Δ +16.2%), accessing money for unexpected (Δ +18.2%) and food-related expenses (Δ +16.8%), self-reported good or very good health (Δ +14.0%), and VLS (Δ +26.7%) were documented. From baseline to endline, significant reductions in experiences with stigma/mistreatment (Δ –12.6%), perceived negative community attitudes towards HIV-positive people (Δ –17.6%), being too sick to participate in daily activities in the past month (Δ –10.2%), and accepting attitudes towards corporeal spousal punishment (Δ –10.6%) were also. Among non-beneficiaries, only significant increases in HIV testing (Δ +4.0%) and VLS in the past 6 months (Δ +20.6%) were observed between survey rounds.

## Modeling effects of ZAMFAM implementation on one-year differences in key outcomes

Table 5 presents unadjusted and adjusted Poisson regression PRRs of key health and socio-economic wellbeing indicators, comparing one-year changes between ZAMFAM beneficiaries and non-beneficiaries. In bivariate analysis, receiving ZAMFAM services was associated with significant reductions in engaging in income-generating activities (PRR = 0.41, 95% Confidence Interval [95% CI]: 0.19–0.89, $p = 0.024$) and eating a smaller meal than desired (PRR = 0.66, 95% CI: 0.45–0.95, $p = 0.026$). Likewise, a significant 6% one-year increase in current ART use (PRR = 1.06, 95% CI: 1.02–1.10, $p = 0.004$) was observed when comparing ZAMFAM beneficiaries to non-beneficiaries. In multivariable analysis, only reductions in the proportion of ALHIV engaging in income-generating activities (Adjusted Prevalence Rate Ratio [aPRR] = 0.44, 95% CI: 0.20–0.99, $p = 0.047$) and improvements in current ART use (aPRR = 1.06, 95% CI: 1.01–1.11, $p = 0.015$) remained significantly associated with receipt of ZAMFAM services. Other indicators with significant one-year differences within each study group were attenuated in regression analysis.

Among caregivers, receiving ZAMFAM services was associated with significant reductions in both HIV-related stigma (aPRR = 0.49, 95% CI: 0.28–0.88, $p = 0.017$) and perceived negative community attitudes towards HIV-positive people (aPRR = 0.77, 95% CI: 0.62–0.96,

**Table 3. Frequency and percent (%) change in key health and socio-economic wellbeing indicators from baseline to endline among adolescents living with HIV (ALHIV), by Zambia Family (ZAMFAM) Project beneficiary status (N = 544).**

| | ZAMFAM Beneficiaries | | | | Non-Beneficiaries | | | |
|---|---|---|---|---|---|---|---|---|
| | Baseline n (%) | Endline n (%) | Δ Post–Pre | p- value | Baseline n (%) | Endline n (%) | Δ Post–Pre | p- value |
| *Social Protection and Psychosocial Wellbeing* | | | | | | | | |
| Has basic social support across four domains | 183 (67.3) | 192 (75.9) | **+8.6%** | **0.029** | 171 (62.9) | 164 (68.1) | +5.2% | 0.219 |
| Any symptoms of depression, past 6 months | 253 (93.0) | 227 (90.1) | −2.9% | 0.226 | 188 (69.1) | 161 (66.8) | −2.3% | 0.575 |
| Experienced stigma or mistreatment because of HIV status | 35 (12.9) | 27 (10.7) | −2.2% | 0.446 | 40 (14.7) | 22 (9.1) | −5.6% | 0.053 |
| Perceives community to harbor negative attitudes about HIV+ people* | 74 (47.4) | 59 (34.5) | **−12.9%** | **0.015** | 71 (43.3) | 70 (44.3) | +1.0% | 0.855 |
| *Education and Labor* | | | | | | | | |
| Consistent school attendance, past 7 days (n = 357) | 118 (63.1) | 125 (70.6) | +7.5% | 0.128 | 119 (70.0) | 114 (68.7) | −1.3% | 0.792 |
| Progressed in school the previous academic year (n = 431) | 163 (78.0) | 155 (74.2) | −3.8% | 0.359 | 152 (68.5) | 150 (70.1) | +1.6% | 0.713 |
| Engaged in any non-household labor for income* | 24 (15.5) | 15 (8.7) | −6.8% | 0.060 | 20 (12.2) | 27 (17.1) | −4.9% | 0.214 |
| *Food and Nutrition* | | | | | | | | |
| Ate a smaller meal than desired at least once, past 4 weeks | 158 (58.1) | 125 (49.4) | **−8.7%** | **0.046** | 89 (32.7) | 71 (29.5) | −3.2% | 0.426 |
| Skipped a meal at least once because food unavailable, past 4 weeks | 86 (31.6) | 67 (26.5) | −5.1% | 0.196 | 56 (20.6) | 64 (26.6) | +6.0% | 0.111 |
| Went to bed hungry at least once, past 4 weeks | 101 (37.1) | 85 (33.6) | −3.5% | 0.397 | 47 (17.3) | 43 (17.8) | +0.5% | 0.867 |
| Went a whole day and evening without eating at least once, past 4 weeks | 48 (17.7) | 37 (14.6) | −3.1% | 0.348 | 26 (9.6) | 21 (8.7) | −0.9% | 0.741 |
| *Physical Health and Wellness* | | | | | | | | |
| Self-reported health status is good or very good | 132 (48.5) | 167 (66.0) | **+17.5** | **<0.001** | 165 (60.7) | 191 (79.3) | **+18.6%** | **<0.001** |
| Too sick to participate in daily activities at least once, past 4 weeks | 91 (33.5) | 74 (29.3) | −4.2% | 0.300 | 83 (30.5) | 54 (22.4) | **−8.1%** | **0.038** |
| Hospitalized for any illness, past 6 months | 30 (11.0) | 19 (7.5) | −3.5% | 0.166 | 28 (10.3) | 18 (7.5) | −2.8% | 0.264 |
| *HIV Clinical Outcomes* | | | | | | | | |
| Currently on ART | 251 (92.3) | 249 (98.4) | **+6.1%** | **0.001** | 264 (97.1) | 235 (97.5) | +0.4% | 0.754 |
| Daily adherence to ART, past 30 days (n = 515) | 222 (88.5) | 227 (91.2) | +2.7% | 0.315 | 230 (87.1) | 209 (88.9) | +1.8% | 0.534 |
| Retained in HIV treatment, past 12 months (n = 514) | 243 (97.2) | 241 (96.8) | −0.4% | 0.787 | 250 (94.7) | 222 (94.5) | −0.2% | 0.910 |
| CD4+ count monitoring, past 6 months | 179 (65.8) | 163 (64.4) | −1.4% | 0.740 | 178 (65.4) | 171 (71.0) | +5.6% | 0.181 |
| Viral load screening, past 6 months | 80 (29.4) | 135 (53.4) | **+24.0%** | **<0.001** | 115 (42.3) | 154 (63.9) | **+21.6%** | **<0.001** |
| *HIV-Related Knowledge, Attitudes, and Behaviors* | | | | | | | | |
| Comprehensive HIV knowledge* | 31 (20.0) | 61 (35.7) | **+15.7%** | **0.002** | 31 (18.9) | 37 (23.4) | +4.5% | 0.321 |
| Ever talked about sex with a household/family member* | 36 (23.2) | 28 (16.4) | −6.8% | 0.120 | 24 (14.6) | 22 (13.9) | −0.7% | 0.856 |
| Ever discussed HIV/AIDS with a household/family member* | 58 (37.4) | 56 (32.8) | −4.6% | 0.377 | 53 (32.2) | 54 (34.2) | +2.0% | 0.723 |

*Only measured among ALHIV aged 10–17 years.

p = 0.018). ZAMFAM service-delivery was additionally associated with significant increases in joint decision-making in how earned money is spent among married caregivers (aPPR = 1.41, 95% CI: 1.04–1.93, p = 0.029); household capacity to access money for unexpected expenses (aPRR = 1.54, 95% CI: 1.17–2.04, p = 0.002) and food-related expenses (aPRR = 1.48, 95% CI: 1.16–1.90, p = 0.002); and self-reported good or very good health status (aPRR = 1.46, 95% CI:

**Table 4. Frequency and percent (%) change in key health and socio-economic wellbeing indicators from baseline to endline among caregivers of adolescents living with HIV, by Zambia Family (ZAMFAM) Project beneficiary status (N = 544).**

| | ZAMFAM Beneficiaries | | | | Non-Beneficiaries | | | |
|---|---|---|---|---|---|---|---|---|
| | Baseline *n* (%) | Endline *n* (%) | Δ Post–Pre | *p*- value | Baseline *n* (%) | Endline *n* (%) | Δ Post–Pre | *p*- value |
| *Social Protection and Psychosocial Wellbeing* | | | | | | | | |
| Has someone to help with daily chores when sick | 231 (84.9) | 234 (92.5) | **+7.6%** | **0.007** | 241 (88.6) | 222 (92.1) | +3.5% | 0.181 |
| Experienced stigma or mistreatment because of HIV status (*n* = 322) | 42 (27.3) | 21 (14.7) | **−12.6%** | **0.008** | 50 (29.8) | 51 (34.0) | +4.2% | 0.418 |
| Perceives community to harbor negative attitudes about HIV + people | 163 (59.9) | 107 (42.3) | **−17.6%** | **<0.001** | 188 (69.1) | 153 (63.5) | −5.6% | 0.178 |
| *Financial Security* | | | | | | | | |
| Can meet child's needs better than other households | 54 (19.9) | 48 (19.0) | −0.9% | 0.799 | 111 (40.8) | 82 (34.0) | −6.8% | 0.113 |
| Joint decision-making with spouse in how earned money is spent (*n* = 326) | 62 (42.8) | 79 (59.0) | **+16.2%** | 0.007 | 91 (50.6) | 83 (50.6) | — | 0.992 |
| Can access money to pay for... | | | | | | | | |
| Any unexpected expenses (*n* = 266) | 59 (41.3) | 72 (59.5) | **+18.2%** | **0.003** | 99 (80.5) | 95 (75.4) | −5.1% | 0.333 |
| Food-related expenses (*n* = 503) | 83 (32.4) | 119 (49.2) | **+16.8%** | **<0.001** | 174 (70.5) | 164 (72.9) | +2.4% | 0.557 |
| Educational expenses (*n* = 427) | 57 (26.8) | 67 (35.1) | +8.3% | 0.070 | 139 (65.0) | 124 (64.3) | −0.7% | 0.882 |
| Household more financially secure than others | 11 (4.1) | 9 (3.6) | −0.5% | 0.771 | 17 (6.3) | 20 (8.3) | +2.0% | 0.371 |
| *Food and Nutrition* | | | | | | | | |
| Went to bed hungry at least once, past 4 weeks | 163 (59.9) | 144 (56.9) | −3.0% | 0.484 | 77 (28.3) | 62 (25.7) | −2.6% | 0.511 |
| Went a whole day and evening without eating at least once, past 4 weeks | 83 (30.5) | 70 (27.7) | −2.8% | 0.473 | 52 (19.1) | 51 (21.2) | +2.1% | 0.564 |
| *Physical Health and Wellness* | | | | | | | | |
| Self-reported health status is good or very good | 105 (38.6) | 133 (52.6) | **+14.0%** | **0.001** | 149 (54.8) | 127 (52.7) | −2.1% | 0.637 |
| Too sick to participate in daily activities at least once, past 4 weeks | 174 (64.0) | 136 (53.8) | **−10.2%** | **0.017** | 130 (47.8) | 114 (47.3) | −0.5% | 0.912 |
| Hospitalized overnight for any illness, past 6 months | 27 (17.5) | 18 (12.6) | −4.9% | 0.235 | 17 (10.1) | 13 (8.7) | −1.4% | 0.658 |
| *HIV Clinical Outcomes* | | | | | | | | |
| Ever tested for HIV | 260 (95.6) | 246 (97.2) | +1.6% | 0.313 | 259 (95.2) | 239 (99.2) | **+4.0%** | **0.009** |
| Currently on ART (*n* = 322) | 150 (97.4) | 143 (100.0) | +2.6% | 0.052 | 163 (97.0) | 148 (98.7) | +1.7% | 0.319 |
| Daily adherence to ART, past 30 days (*n* = 313) | 136 (90.7) | 134 (93.7) | +3.0% | 0.334 | 154 (94.5) | 139 (93.9) | −0.6% | 0.833 |
| Retained in HIV treatment, past 12 months (*n* = 313) | 143 (95.3) | 139 (97.2) | +1.9% | 0.400 | 155 (95.1) | 139 (93.9) | −1.2% | 0.650 |
| CD4+ count monitoring, past 6 months (*n* = 322) | 121 (78.6) | 116 (81.1) | +2.5% | 0.585 | 124 (73.8) | 119 (79.3) | +5.5% | 0.247 |
| Viral load screening, past 6 months (*n* = 322) | 73 (47.4) | 106 (74.1) | **+26.7%** | **<0.001** | 83 (49.4) | 105 (70.0) | **+20.6%** | **<0.001** |
| *HIV-Related Knowledge, Attitudes, and Perceptions* | | | | | | | | |
| Comprehensive HIV knowledge | 133 (48.9) | 139 (54.9) | +6.0% | 0.167 | 106 (39.0) | 96 (39.8) | +0.8% | 0.842 |
| Supports harsh physical punishment at school or in the household | 63 (23.2) | 60 (23.7) | +0.5% | 0.881 | 52 (19.2) | 41 (17.0) | −2.2% | 0.537 |
| Agrees beating wife acceptable in at least one of five situations | 129 (47.4) | 93 (36.8) | **−10.6%** | **0.013** | 105 (38.6) | 83 (34.4) | −4.2% | 0.329 |

1.14–1.87, *p* = 0.003). While significant reductions in the proportion of caregivers being too sick to participate in daily activities in the past month and acquiescence towards spousal violence, respectively, were observed, these reductions were not significantly larger among ZAMFAM-beneficiary caregivers compared to non-beneficiaries.

**Table 5. One-year change comparing ZAMFAM beneficiaries to non-beneficiaries Poisson prevalence rate ratios (PRRs), 95% confidence intervals (95% CI), and p-values for ZAMFAM key indicators among adolescents living with HIV (ALHIV) and their caregivers (N = 544).**

| | Unadjusted | | | Adjusted[§] | | |
|---|---|---|---|---|---|---|
| | PRR | 95% CI | p-value | aPRR | 95% CI | p-value |
| *ALHIV* | | | | | | |
| Has basic social support across four domains | 1.04 | 0.89–1.21 | 0.602 | 1.04 | 0.88–1.22 | 0.658 |
| Perceives community to harbor negative attitudes about HIV+ people (n = 355) | 0.70 | 0.50–1.00 | 0.052 | 0.73 | 0.50–1.04 | 0.084 |
| Engaged in any non-household labor for income (n = 356) | **0.41** | **0.19–0.89** | **0.024** | **0.44** | **0.20–0.99** | **0.047** |
| Ate a smaller meal than desired at least once, past 4 weeks | **0.66** | **0.45–0.95** | **0.026** | 0.99 | 0.73–1.34 | 0.934 |
| Self-reported health status is good or very good | 1.04 | 0.87–1.25 | 0.648 | 1.03 | 0.85–1.25 | 0.752 |
| Too sick to participate in daily activities at least once, past 4 weeks | 1.19 | 0.84–1.68 | 0.340 | 1.28 | 0.87–1.89 | 0.207 |
| Currently on ART | **1.06** | **1.02–1.10** | **0.004** | **1.06** | **1.01–1.11** | **0.015** |
| Viral load screening, past 6 months | 1.19 | 0.93–1.53 | 0.164 | 1.22 | 0.93–1.61 | 0.147 |
| Comprehensive HIV knowledge (n = 355) | 1.47 | 0.86–2.52 | 0.161 | 1.41 | 0.81–2.46 | 0.221 |
| *Caregivers* | | | | | | |
| Has someone to help with daily chores when sick | 1.05 | 0.97–1.13 | 0.264 | 1.03 | 0.95–1.21 | 0.464 |
| Experienced stigma or mistreatment because of HIV status (n = 344) | **0.47** | **0.29–0.76** | **0.002** | **0.49** | **0.28–0.88** | **0.017** |
| Perceives community to harbor negative attitudes about HIV+ people | **0.77** | **0.63–0.94** | **0.009** | **0.77** | **0.62–0.96** | **0.018** |
| Joint decision-making with spouse in how earned money is spent (n = 367) | **1.40** | **1.05–1.87** | **0.021** | **1.41** | **1.04–1.93** | **0.029** |
| Can access money to pay for. . . | | | | | | |
| Any unexpected expenses (n = 373) | **1.55** | **1.19–2.02** | **0.001** | **1.54** | **1.17–2.04** | **0.002** |
| Food-related expenses (n = 533) | **1.47** | **1.16–1.86** | **0.002** | **1.48** | **1.16–1.90** | **0.002** |
| Self-reported health status is good or very good | **1.42** | **1.12–1.79** | **0.003** | **1.46** | **1.14–1.87** | **0.003** |
| Too sick to participate in daily activities at least once, past 4 weeks | 0.85 | 0.69–1.05 | 0.128 | 0.86 | 0.68–1.08 | 0.197 |
| Ever tested for HIV | 0.98 | 0.94–1.01 | 0.221 | 0.97 | 0.93–1.01 | 0.180 |
| Viral load screening, past 6 months (n = 344) | 1.10 | 0.86–1.39 | 0.446 | 1.11 | 0.85–1.46 | 0.430 |
| Agrees beating wife acceptable in at least one of five situations | 0.86 | 0.66–1.12 | 0.269 | 0.83 | 0.61–1.13 | 0.241 |

[§] Adjusted for continuous age, sex, province, marital status (caregiver only), disability, household shelter protection, agricultural land and livestock ownership, and survey round.

## Discussion

Findings from this prospective cohort study suggest that ZAMFAM's community-based, multi-component intervention components may positively improve ALHIV health outcomes and strengthen caregiver psychosocial and economic resilience. During the one-year observation period, significant positive changes in caregivers' physical and psychosocial wellbeing as well as financial capacity were observed in ZAMFAM implementation sites. Significant improvements in ALHIV health service use, specifically ART coverage, were also recorded among ZAMFAM beneficiaries; however, many of these changes were accompanied by similar improvements in non-intervention, comparison sites, ultimately attenuating the significance of some intervention effect estimates. As many ZAMFAM intervention components interface directly with caregivers (e.g., parenting skills, loans and savings groups), potential benefits may manifest more quickly among the most proximal program recipients (i.e., caregivers), taking more time for these effects to cascade down to indirect, distal beneficiaries, including ALHIV. Additional rounds of data collection may be needed to estimate the magnitude and significance of sustained intervention impacts on beneficiaries over time.

A noteworthy finding of the study is the immediacy of potential intervention effects on caregiver wellbeing and household economic strengthening, specifically financial security.

A significantly higher proportion of ZAMFAM-beneficiary caregivers, compared to non-beneficiaries, reported increased access to money for unexpected and food-related expenses, respectively, as well as joint decision-making in how earned money is spent, over the one-year study period. Additionally, significant reductions in labor force participation among ZAMFAM-beneficiary ALHIV could reflect increased financial security in households receiving ZAMFAM's overlapping economic strengthening interventions, specifically community savings groups and agricultural subsidies. Complementing findings from longitudinal assessments of similar community-based integrated service-delivery initiatives [19, 25, 32], these results reinforce the importance of economic strengthening interventions directly engaging with and supporting the caregivers of ALHIV and other vulnerable children.

Promising improvements in social protection and psychosocial wellbeing, particularly with respect to HIV-related stigma and perceived community attitudes, among caregivers underscore potential effectiveness of ZAMFAM's community-based psychosocial counseling and parenting interventions. Noteworthy reductions in HIV-related stigma and perceived negative community attitudes towards HIV-positive people among caregivers receiving ZAMFAM services were observed. While these reductions were not significantly different comparing ZAMFAM-beneficiary ALHIV to non-beneficiaries, sizeable declines, nonetheless, suggests improvements in psychosocial status and resilience of ZAMFAM-beneficiary households. With more prolonged exposure to ZAMFAM activities and a larger study sample, the magnitude of these differences may have been larger and significant. These results, consistent with findings from existing literature, underscore the contribution of psychosocial support interventions to stigma mitigation, increased social support, and resilience-building, particularly for young people [22–25, 32, 33].

With the noteworthy exception of ART coverage, ZAMFAM service-delivery was not significantly associated with changes in HIV care and treatment outcomes. Self-reported ART adherence and retention in HIV care among ALHIV remained high (over 90%) throughout the study period. These care and treatment estimates, among beneficiaries and non-beneficiaries alike, are substantially higher than what is observed in population-based seroprevalence studies in Zambia, where fewer than half (53.1%) of children (<15 years) and three-fourths (71.3%) of adolescents (15–24 years), respectively, are virally suppressed [3]. As such, this mitigated power to detect significant changes in HIV care and treatment outcomes in a sample with high ART adherence and retention at baseline. Sustained trends in HIV care engagement and retention, nonetheless, underscore potential effectiveness of ZAMFAM and other HIV interventions in supporting the most vulnerable ALHIV. Furthermore, significant improvements in VLS coverage among ZAMFAM beneficiaries and non-beneficiaries alike could be attributed to secular effects of VLS scale-up efforts in Zambia, beginning in 2017 [27]. Irrespective of the null effect of ZAMFAM implementation on VLS coverage documented in this study, sustaining gains in VLS scale-up across intervention contexts is needed to improve clinical monitoring of ALHIV, support differentiated service-delivery efforts, and build case-based HIV surveillance capacity.

Food security, illness, and schooling continued to present formidable challenges for ALHIV and caregivers, with few observed changes between survey rounds. While some reductions were observed in household food security, nutrition, and illness indicators for ALHIV, many of these differences were not significant between baseline and endline. Furthermore, none of the prevalence differences for these indicators in the one-year assessment period were significantly different for ZAMFAM beneficiaries than non-beneficiaries. School attrition and progression estimates among ZAMFAM beneficiaries declined during the study's assessment, though these changes could be attributed to other vulnerabilities and speak less to the

ineffectiveness of ZAMFAM programming in improving school attendance. School attrition in Zambia is highest following completion of primary school (prior to secondary school) [34]; because adolescents included in this study are in the age range where school dropout is most likely to occur, the risk of school attrition during the study period is high and, therefore, observing this outcome in this cohort is unsurprising. Additional research can help unpack predictors of school attrition among ZAMFAM beneficiaries and explore reasons why ZAM-FAM services may have positive or null effects on school retention.

## Limitations

As this study employed an observational, non-randomized prospective design, attribution of observed changes in key health and socio-economic outcomes to ZAMFAM interventions is limited, as causal relationships cannot be inferred. Because the study compared numerous outcomes in cohort participants receiving and not receiving ZAMFAM services, observed relationships between primary outcomes and ZAMFAM implementation cannot be attributed to specific, individual interventions within the ZAMFAM service package. Furthermore, information about intervention coverage were not collected as part of this study, restricting investigation of dose-response relationships between intervention frequency and key outcomes. Further research is needed to interrogate which combination(s) of ZAMFAM services and intervention dosage(s) optimize health and wellbeing for ALHIV and their caregivers. Different sampling strategies for recruiting a community-based sample of ZAMFAM beneficiaries (in Central Province) and clinical population of non-beneficiaries (in Eastern Province) resulted in heterogenous study groups, limiting their comparability and potentially attenuating capacity to detect significant changes between study groups over time. Because non-beneficiary households from Eastern Province were recruited from HIV clinics (rather than in community settings, like ZAMFAM beneficiaries), ALHIV and caregivers in these comparison households may differed substantially on key indicators (e.g., HIV care and treatment outcomes, financial insecurity, psychosocial wellbeing) than households not accessing HIV services. Additionally, comparison districts in Eastern Province from which non-ZAMFAM beneficiaries were recruited were not matched to intervention districts in Central Province on other salient compositional factors, included socio-economic status or demographic factors. The moderately sized sample, coupled with a study observation period limited to one year, may have further restricted capacity to detect programmatic and temporal effects of ZAMFAM implementation on key outcomes. Given data were self-reported by ALHIV and their caregivers, inferences drawn from these results should consider the potential contributions acquiescence and recall biases to study findings. Finally, while insights derived from this study may be applicable to OVC and ALHIV programming in sub-Saharan Africa, results should be conservatively generalized to populations and settings beyond those described in this study.

## Conclusions

Implementation of a multi-component, community-based, integrated service-delivery package was associated with significant improvements in financial security and wellbeing of caregivers of ALHIV in Zambia. Within the short one-year assessment window, marginal improvements in the health status of ALHIV, specifically ART coverage, were also observed. Future research should examine service delivery-related outcomes (e.g., fidelity to process and structure, program acceptability and coverage, cost and feasibility) to better understand how findings from this study are related to the processes and mechanics of ZAMFAM implementation. Measurement of these implementation-related outcomes is needed to ascertain the ZAMFAM project's

scalability and sustainability. To improve outcomes in ALHIV, ZAMFAM implementers should strengthen community-based delivery of psychosocial and health support to ALHIV, particularly during episodes of illness and other care-destabilizing events, leveraging existing service-delivery infrastructure to identify and monitor the most vulnerable households. Sustaining gains in viral load screening, both in ZAMFAM catchment areas and non-intervention sites, will be essential for clinical decision-making and monitoring progress towards HIV epidemic control.

## Supporting information

**S1 Table. Measures of internal consistency (Cronbach's alpha) for social protection and psychosocial wellbeing constructs derived from MEASURE Evaluation's OVC Survey Toolkit for adolescents living with HIV (ALHIV) and their adult caregivers at baseline (*N* = 544).**
(DOCX)

## Acknowledgments

We thank the participants and data collectors for their support of and participation in the study. We would like to thank the implementing partner of ZAMFAM in Central Province, Development Aid from People to People (DAPP), for their support, cooperation, and guidance during data collection. We are further grateful to the Government of the Republic of Zambia, through the Ministry of Health and the Ministry of Community Development and Social Services, for approving the study and for availing the services of their staff in the various districts where the study was conducted.

## Author Contributions

**Conceptualization:** Michael T. Mbizvo, Nkomba Kayeyi.

**Data curation:** Joseph G. Rosen, Lyson Phiri, Mwelwa Chibuye, Edith S. Namukonda.

**Formal analysis:** Joseph G. Rosen, Lyson Phiri, Mwelwa Chibuye.

**Funding acquisition:** Michael T. Mbizvo, Nkomba Kayeyi.

**Project administration:** Lyson Phiri, Mwelwa Chibuye.

**Supervision:** Michael T. Mbizvo, Nkomba Kayeyi.

**Writing – original draft:** Joseph G. Rosen.

**Writing – review & editing:** Lyson Phiri, Mwelwa Chibuye, Edith S. Namukonda, Michael T. Mbizvo, Nkomba Kayeyi.

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
