## [Decision Letter · Decision Letter 0]

16 Sep 2020

PONE-D-20-22311

Integrated psychosocial, economic strengthening, and clinical service-delivery to improve health and resilience of young people living with HIV and their caregivers: Findings from a prospective cohort study in Zambia

PLOS ONE

Dear Dr. Rosen,

Thank you for submitting your manuscript to PLOS ONE. After careful consideration, we feel that it has merit but does not fully meet PLOS ONE’s publication criteria as it currently stands. Therefore, we invite you to submit a revised version of the manuscript that addresses the points raised during the review process.

We look forward to receiving your revised manuscript.

Kind regards,

Joel Msafiri Francis, MD, MS, PhD

Academic Editor

PLOS ONE

Journal Requirements:

2. Please include additional information regarding the survey or questionnaire used in the study and ensure that you have provided sufficient details that others could replicate the analyses.

For instance, if you developed a questionnaire as part of this study and it is not under a copyright more restrictive than CC-BY, please include a copy, in both the original language and English, as Supporting Information.

Reviewers' comments:

Reviewer's Responses to Questions

**Comments to the Author**

1. Is the manuscript technically sound, and do the data support the conclusions?

Reviewer #1: Yes

Reviewer #2: Yes

Reviewer #3: Yes

2. Has the statistical analysis been performed appropriately and rigorously? 

Reviewer #1: Yes

Reviewer #2: Yes

Reviewer #3: Yes

3. Have the authors made all data underlying the findings in their manuscript fully available?

Reviewer #1: Yes

Reviewer #2: Yes

Reviewer #3: Yes

4. Is the manuscript presented in an intelligible fashion and written in standard English?

Reviewer #1: Yes

Reviewer #2: Yes

Reviewer #3: Yes

5. Review Comments to the Author

Reviewer #1: Important note: This review pertains only to ‘statistical aspects’ of the study and so ‘clinical aspects’ [like medical importance, relevance of the study, ‘clinical significance and implication(s)’ of the whole study, etc.] are to be evaluated [should be assessed] separately/independently. Further please note that any ‘statistical review’ is generally done under the assumption that (such) study specific methodological [as well as execution] issues are perfectly taken care of by the investigator(s). This review is not an exception to that and so does not cover clinical aspects {however, seldom comments are made only if those issues are intimately / scientifically related & intermingle with ‘statistical aspects’ of the study}. Agreed that ‘statistical methods’ are used as just tools here, however, they are vital part of methodology [and so should be given due importance].

COMMENTS: Basically study on young people living with HIV in country like Zambia is most welcome, important and appreciable. Moreover, this study is very well planned (with correct methodology, etc.), however, I have few doubts:

- According to statement made on page 11, [“At endline (July–September 2018), study enumerators returned to households sampled at baseline for a follow-up interview using similar data collection materials”] end line survey was conducted during July–September 2018, then why this delay in publication?

- Execution of such community trials is generally ‘critical’ [mostly because ‘no control’ on comparison group/sample] and has to be undertaken more carefully. It is desirable to discuss particular/specific steps taken [to ensure valid data / data quality, etc]. They are not discussed adequately, in my opinion.

- Footnote of table-2 gives classification/categorization of P-vales [a p < 0.05; b p < 0.01; c p < 0.001] but is not used {in fact indicating this is not necessary as actual P-values are reported}.

- Although data analyses done are correct but whether could have been performed by simple methods [for example, data of table-3 analysed by Poisson regression with generalized estimating equations (reference 29 quoted is also excellent) but could have done {why make all ‘binary’, few could be measured in scale} by Wilcoxon’s Signed Ranked test (not Wilcoxon rank-sum test which is incidentally same as well-known Mann-Whitney test) or for ‘binary data by McNemar’s test]? In my opinion, analyses should be as simple as possible. Statistical techniques used simply for better understanding of your observations only, I guess.

With the backdrop of ‘limitations’ described on page 27, best possible work is done which is indeed appreciable/acceptable.

Reviewer #2: HIV care and treatment. Program implementers convene health staff and community health

workers (CHWs), identifying strategies for scaling HIV testing as well as viral load screening (VLS)

and CD4+ count monitoring for those living with HIV. CHWs are additionally paired with YPLHIV

to monitor and support adherence to care and treatment.

Comment: What kind of strategies for scaling HIV testing were implemented here? Were the strategies uniform throughout the households?

Parenting. Contracting with a local women’s religious organization, program implementers train

community staff on parenting skills, including child abuse and gender-based violence.

Community staff subsequently mobilize caregivers of YPLHIV and hold meetings to discuss these

issues.

Cooment: How did this religious organization reach out households of non-believers? can you explain how study mitigated bias or untoward effects of using such organization with uptake of this initiative?

Food security. Farming inputs – including maize seed, legumes, cassava, sweet potatoes,

chickens, and goats – are provided to OVC and YPLHIV households.

Comment: How did intervention contribute towards results in Table 3 under Food and Nutrition?

Household economic strengthening. Loans and savings schemes are introduced into Village

Action Groups, allowing for members to borrow money for investments (e.g., in businesses,

essential household items) or purchase agricultural inputs at lower interest rates.

Comment:Did you provide cash as part of intervention for study participants to invest? if yes, how did you evaluate the project to see if the intervention worked?

Psychosocial support. YPLHIV are grouped with their primary caregiver and a neighbor to

support the child’s adherence to ART and engagement with HIV services. Counselors and CHWs

additionally receive specialized training to enhance their counseling skills

Comment: what do you mean a by being paired up by a neighbor? How did you handle disclosure issues here?

Four matched districts in Eastern Province – Nyimba, Chipata, Petauke, and Lundazi – were selected as

non-intervention, comparison sites based on HIV prevalence estimates for children under 15 from 2016-

17 Health Management Information System records of new HIV infections in selected districts as well as

the districts’ geographic positioning along main thoroughfares

Comment:Did you look at other factors such as socio economic status of matching districts?

Was there any spilling effect geographically?

Table 1. Operationalization of key Zambia Family (ZAMFAM) Project outcome variables

comment: What criteria did you use to categorize 'definition' into each domain? For instance how did Depressive symptoms get aggregated into a single domain?

How did you know definition of certain outcome truly define it? did you measure the internal consistency?

Table 3-YPLHIV

Comment: Is it possible to disaggregrate your findings by age group with interest to variations in the younger population particularly children and youth to see how the change in key health and socio-economic well-being

indicators had taken place after the intervention? The idea is to see if this intervention truly reached the intended population

General comment: Can this intervention be scaled up to your country? and other similar settings?

Reviewer #3: The authors in this article seek to evaluate the impact of an integrated HIV intervention on the health and wellbeing of adolescents living with HIV and their caregivers in Zambia. They use a prospective cohort study design, and compare outcomes among participants in the intervention versus non-intervention areas as well as assess changes in these outcomes a year later. The intervention evaluated is a PEPFAR funded programme (ZAMFAM) that targeted five areas (HIV care and treatment, parenting, food security, household economic strengthening, psycho-social support).

Overall, this is paper is well written. I commend the authors for going beyond narrow HIV treatment outcomes and examining the effects of broader indicators.

I approve for publication with minor revisions (detailed below)

Introduction

- Kindly update guidelines to reflect latest UNAIDS goal ( 95-95-95 ).

- I would recommend highlighting why these multi-component intervention and evaluations assessing health and wellbeing outcomes are important in the context of the SDGs.

- Could the author describe current HIV cascade outcomes among YPLHIV and in the discussion relate findings to this?

- There is a word missing from this sentence … “as these compound adversities can destabilize access to and retention in HIV services”

- Perhaps consider more standard terminology for the age-range under investigation. See UNAIDS definition on YPLHIV (it is aged 15-24 years). A more appropriate definition that is aligned to the study age range is adolescents living with HIV (ALHIV).

Methods

- The authors highlight the theory of change of the programme. It would be good to include a model/figure highlighting key hypothesised pathways.

- Definitions and measures of wellbeing have been noted in the HIV literature. The authors should highlight how this construct was conceptualised in their study and why. In addition, the term resilience is used with regards to the caregiver and environment. It would be important to define this construct.

- Misuse of the comma: “Dovetailing existing, nationally scaled interventions for OVC and YPLHIV in Zambia…”

- The authors in some sections describe the intervention as being “multi-sectoral”. I would suggest expanding on this. Which sectors were involved and how was the intervention implemented?

- The authors indicate “Importantly, comparison sites in Eastern Province did not have similar or comparable interventions to ZAMFAM at the time of study enrollment”. However, it is unclear how this was assessed.

- Were measures of depression and stigma based on validated measures? If not, the authors should indicate this /highlight this under limitations.

- Data collection- it is unclear who administered the baseline and follow-up questionnaire. Were these CHWs that ran the intervention or an external group of fieldstaff not involved in intervention delivery?

- The authors should describe the reason for this modelling approach.

Results

- What % of beneficiaries and non-beneficiaries resided in urban vs. rural areas?

- As mentioned above, it would be important to highlight what informed the selection of indicators for categories in Table 3 (e.g. social protection and psycho-social wellbeing). Stigma, depression and social support are regarded as more correlates of wellbeing.

- Table 4 does not include the variable on depressive symptoms- was this not probed in the caregiver questionnaire?

-

Discussion

- Could the authors describe why receipt of ZAMFAM services was associated with decreases in the % of YPLHIV engaging in income generating activities?

- The literature indicates positive spillovers with regards to caregiver wellbeing. Could the authors explain why the intervention was associated with wellbeing benefits for the caregiver and not YPLHIV?

- Furthermore, could the authors explain why the intervention was not associated with positive HIV treatment and care effects for YPLHIV?

- Could authors explain further what other sources could have resulted in the declines in school retention among beneficiaries?

- To help unpack the above, could the authors also highlight information regarding intervention dose? What % of caregivers or YPLHIV attended sessions related to intervention activities.

- Limitations- it would be important to highlight the impact of selection bias on the direction of study findings. It is unclear what is meant by the term “differential sampling”. The authors should discuss the generalizability of study findings.

- A statement on future research is needed- particularly with regards to other implementation science outcome measures (feasibility, acceptability, costs).

6. PLOS authors have the option to publish the peer review history of their article (what does this mean?). If published, this will include your full peer review and any attached files.

Reviewer #1: No

Reviewer #2: No

Reviewer #3: No

---

## [Author Response · Author response to Decision Letter 0]

31 Oct 2020

We thank the editors and reviewers for their thorough, detailed review of our manuscript. We have included a point-by-point response to each comment in a separate document, submitted with our manuscript files.

---

## [Decision Letter · Decision Letter 1]

27 Nov 2020

Integrated psychosocial, economic strengthening, and clinical service-delivery to improve health and resilience of adolescents living with HIV and their caregivers: Findings from a prospective cohort study in Zambia

PONE-D-20-22311R1

Dear Dr. Rosen,

We’re pleased to inform you that your manuscript has been judged scientifically suitable for publication and will be formally accepted for publication once it meets all outstanding technical requirements.

Kind regards,

Joel Msafiri Francis, MD, MS, PhD

Academic Editor

PLOS ONE

Additional Editor Comments (optional):

Reviewers' comments:

Reviewer's Responses to Questions

**Comments to the Author**

1. If the authors have adequately addressed your comments raised in a previous round of review and you feel that this manuscript is now acceptable for publication, you may indicate that here to bypass the “Comments to the Author” section, enter your conflict of interest statement in the “Confidential to Editor” section, and submit your "Accept" recommendation.

Reviewer #1: All comments have been addressed

Reviewer #2: (No Response)

Reviewer #3: All comments have been addressed

2. Is the manuscript technically sound, and do the data support the conclusions?

Reviewer #1: Yes

Reviewer #2: Yes

Reviewer #3: Yes

3. Has the statistical analysis been performed appropriately and rigorously? 

Reviewer #1: Yes

Reviewer #2: Yes

Reviewer #3: Yes

4. Have the authors made all data underlying the findings in their manuscript fully available?

Reviewer #1: Yes

Reviewer #2: Yes

Reviewer #3: No

5. Is the manuscript presented in an intelligible fashion and written in standard English?

Reviewer #1: Yes

Reviewer #2: Yes

Reviewer #3: Yes

6. Review Comments to the Author

Reviewer #1: Since the comments made on earlier draft by me (and hopefully by other respected reviewers also) are attended positively/adequately, now the manuscript is improved a lot. No major issue left, in my opinion.

Reviewer #2: I have no further comments to authors.

Reviewer #3: (No Response)

7. PLOS authors have the option to publish the peer review history of their article (what does this mean?). If published, this will include your full peer review and any attached files.

Reviewer #1: No

Reviewer #2: **Yes: **Theodora Mbunda

Reviewer #3: No

---

## [Editor Report · Acceptance letter]

13 Jan 2021

PONE-D-20-22311R1 

Integrated psychosocial, economic strengthening, and clinical service-delivery to improve health and resilience of adolescents living with HIV and their caregivers: Findings from a prospective cohort study in Zambia 

Dear Dr. Rosen:

I'm pleased to inform you that your manuscript has been deemed suitable for publication in PLOS ONE. Congratulations! Your manuscript is now with our production department. 

Kind regards, 

on behalf of

Dr. Joel Msafiri Francis 

Academic Editor

PLOS ONE